# Plant Extracts from the Yucatan Peninsula in the In Vitro Control of *Curvularia lunata* and Antifungal Effect of *Mosannona depressa* and *Piper neesianum* Extracts on Postharvest Fruits of Habanero Pepper

**DOI:** 10.3390/plants12162908

**Published:** 2023-08-09

**Authors:** Patricia Cruz-Cerino, Jairo Cristóbal-Alejo, Violeta Ruiz-Carrera, Marcela Gamboa-Angulo

**Affiliations:** 1Unidad de Biotecnología, Centro de Investigación Científica de Yucatán, Merida 97205, Mexico; patriciacruzcerino@gmail.com; 2Laboratorio de Fitopatología, Tecnológico Nacional de México, Campus Conkal, Conkal 97345, Mexico; 3División Académica de Ciencias Biológicas, Universidad Juárez Autónoma de Tabasco, Villahermosa 86039, Mexico; violeta@ujat.mx

**Keywords:** antifungal, α-asarone, *Capsicum chinense*, plant extracts, *Mosannona depressa*, *Piper neesianum*

## Abstract

Plant extracts are a valuable alternative for the control of phytopathogenic fungi in horticultural crops. In the present work, the in vitro antifungal effect of ethanol and aqueous extracts from different vegetative parts of 40 native plants of the Yucatan Peninsula on *Curvularia lunata* ITC26, a pathogen of habanero pepper (*Capsicum chinense*), and effects of the most active extracts on postharvest fruits were investigated. Among these, the ethanol extracts of *Mosannona depressa* (bark from stems and roots) and *Piper neesianum* (leaves) inhibited 100% of the mycelial growth of *C. lunata*. The three extracts were partitioned between acetonitrile and *n*-hexane. The acetonitrile fraction from *M. depressa* stem bark showed the lowest mean inhibitory concentration (IC_50_) of 188 µg/mL against *C. lunata*. The application of this extract and its active principle α-asarone in the postharvest fruits of *C. chinense* (500 µg/mL) was shown to inhibit 100% of the severity of the infection caused by *C. lunata* after 11 days of contact. Both samples caused the distortion and collapse of the conidia of the phytopathogen when observed using electron microscopy at 96 h. The spectrum of *M. depressa* enriched antifungal action is a potential candidate to be a botanical fungicide in the control of *C. lunata* in cultivating habanero pepper.

## 1. Introduction

The species of the genus *Capsicum* spp. (pepper) are among the most appreciated vegetables worldwide with an annual production of 3,112,480 tons in Mexico, meaning that it is considered to rank second [1]. Among the *Capsicum* species, the habanero pepper (*Capsicum chinense* Jacq.) is mainly cultivated in the states of Campeche, Yucatan, and Quintana Roo (25,128 tons in 2022), which is appreciated worldwide for its spiciness due to its capsaicin content that has diverse applications in medicine and biotechnology [2,3].

Plant pathogenic fungi continuously threaten plant production; anthracnose and fruit rot of *Capsicum* spp. Are two of the main postharvest diseases caused by *Colletotrichum* spp., including *Colletotrichum acutatum*, *C. scovillei*, *C. gloeosporioides*, *C. truncatum*, [4,5,6] and others such as *Curvularia lunata* [7,8], representing one of the major problems faced when marketing high-quality fruits. In the state of Yucatán, *Capsicum* spp. horticultural production is limited by the presence of fungal diseases (leaf spot and fruit necrosis) identified as *Alternaria alternata*, *A. brassicicola*, *A. solani*, *C. lunata*, *Colletotrichum capsici*, *C. truncatum*, *Helminthosporuim* sp., and *Penicillium oxalicum* [9,10,11,12]. Among these phytopathogens, *C. lunata* is associated with the postharvest fruit of the habanero pepper, which is one of the most destructive pathogens responsible for stem blight, leaf spot, leaf blight, root rot, and necrotic rot in banana, rice, and spinach [13,14,15]; it has been little explored in the habanero pepper fruit.

The foliar and postharvest disease control of *C. lunata* has been carried out via the intensive use of synthetic fungicides, such as benomyl, carbendazim, imazalil, imidacloroprid, hexaconazole, mancozeb, and tebuconazole [8,12,16,17]. Unfortunately, the inappropriate application of these products results in resistance and collateral environmental effects on ecosystems, beneficial soil organisms, insects, pollinators, other organisms, and humans [18,19]. Agrochemical products of plant origin are nowadays evaluated more intensively to offer alternative products to farmers to move towards more sustainable production and reduce the risk of resistance development in pathogens [20,21,22,23]. Few studies have demonstrated the fungicidal potential of plant extracts in the control of the postharvest pathogens of *Capsicum* spp. fruits, such as 1% oil from *Azadirachta indica* leaves, 20% aqueous extract from *Lantana camara* leaves, and ethanolic extracts from *Abrus precatorius* and *Rauvolfia tetraphylla* roots [8,24,25].

To provide new antifungal products, our working group has monitored the extracts of plants native to the biotic peninsula of Yucatan against fungal phytopathogens that affect various agricultural or ornamental crops. Among these, effective extracts are the aqueous extracts from *Bonellia flammea* stem bark that inhibited the mycelial growth on the *C. lunata* strains ITC26, ITC22, and ITC4 isolated from *C. chinense* [9], *Solanum lycopersicum* [26], and *Thrinax radiate* [27], respectively, and *Croton chichenensis* root extract, which inhibited *C. lunata* ITC22 [26]. The ethanolic extract from *Acalypha gaumeri* roots has also been reported to inhibit *C. lunata* ITC10 isolated from *Zea mays* L. [28]. In vivo, an aqueous extract of *B. flammea* stem bark effectively controlled *C. lunata* and *P. oxalicum* fungi on *C. chinense* fruits [9]. Continuing the bioprospecting search for natural plant agrochemical options to control plant pathogens and parasites, our group collected a second and larger number of species from six sites in the Yucatan Peninsula. The criteria for selecting plant species (40) were local availability, chemotaxonomical and ethnobotanical antecedents, absence of environmental risk restrictions, and some at serendipity.

Research with plant extracts for the in vivo control of postharvest diseases caused by *C. lunata* on habanero peppers is limited. In the present study, the in vitro antifungal activity of aqueous and ethanolic extracts from 40 plants of the Yucatan Peninsula against the phytopathogen *C. lunata* ITC26 is reported. The effect of the most active extracts on hyphal morphology was estimated using scanning electron microscopy (SEM), and the in vivo control of *C. lunata* infection in the postharvest fruits of habanero pepper was determined.

## 2. Results

### 2.1. In Vitro Activity of Plant Extracts on Curvularia lunata

The results of the screening of species effective against *C. lunata* ITC26 are presented in Table 1 In total, 13 extracts were active against *C. lunata* at 2000 µg/mL, corresponding to eight ethanolic extracts from *Alvaradoa amorphoides* leaves, *Licaria* sp. roots., *Helicteres baruensis* leaves and stems, *Mosannona depressa* stem bark and root bark, and *Piper neesianum* leaves and roots. The most active were the ethanolic extracts from *Mosannona depressa* stem bark and root bark and *Piper neesianum* leaves with lethal effects on *C. lunata*. In contrast, the extracts from the leaves of *A. amorphoides*, *Licaria* sp., and *H. baruensis* and the roots of *P. neesianum* moderately affected (MCI = 75%) the mycelial growth of this pathogen (Table 1). Only five aqueous extracts showed a significant capacity to inhibit the growth of *C. lunata* (MCI = 25%) at a concentration of 3% (*w*/*v*), corresponding to the extracts of *Byrsonima bucidifolia* leaves, stems, and roots and *Morella cerifera* leaves and roots.

### 2.2. Determination of Minimum Inhibitory Concentration of Extracts, Fractions, and α-Asarone

The three most active extracts from *M. depressa* and *P. neesianum* were partitioned, and their minimum inhibitory concentration (MIC) against *C. lunata* ITC26 was determined. The results showed that the lowest MIC was 250 µg/mL caused by the ethanolic extracts of stem bark and root bark from *M. depressa*, both with fungicidal effects. With lower activity and a MIC of 500 µg/mL, the precipitate of *M. depressa* root bark, the ethanolic extract from *P. neesianum* leaves, and its acetonitrile fraction were detected as well as the standard commercial α-asarone, all with fungistatic effects. The hexane fractions from the three active ethanolic extracts were inactive against *C. lunata* ITC26 (Table 2).

### 2.3. Inhibitory Concentration (IC_50_ and IC_95_) of the Most Active Extracts, Fractions, and α-Asarone

The ethanolic extracts from *M. depressa* stem bark showed the lowest IC_50_ and IC_95_ of 188 and 218 µg/mL, respectively, against *C. lunata* (Table 3). The α-asarone showed an IC_50_ of 190 µg/mL, which is equivalent to the ethanolic extracts, but its IC_95_ of 325 µg/mL was higher. The IC_50_ and IC_95_ for the precipitate from *M. depressa* root bark (229 and 265 µg/mL, respectively) and *P. neesianum* acetonitrile (222 and 256 µg/mL, respectively) fractions were equivalent against *C. lunata* (Table 3).

### 2.4. Effect of Active Extracts on Curvularia lunata

Exposure of *C. lunata* hyphae and conidia to the negative control for 96 h confirmed the well-formed hyphae and ovoid-shaped conidia characteristics of the fungal species (Figure 1A–C). After 96 h exposure to ethanolic extracts from *M. depressa* bark from the stems and roots at 2000 µg/mL as well as to standard α-asarone at 500 µg/mL, dehydrated and contorted hyphae and fully dry conidia were observed (Figure 1D–F).

### 2.5. In Vivo Effect of Extracts and α-Asarone against C. lunata on Habanero Pepper Fruits

The analysis of variance to estimate the effect of the treatments on the decrease in severity or control of *C. lunata* isolated from postharvest fruits showed significant statistical differences (*p* ≤ 0.01) among the treatments (Table 4, Figure 2). Treatments T1, T2, T4, T7, and T10 had a lethal effect (IC = 100%) on the control of *C. lunata* infection in the postharvest fruits of *C. chinense* 11 days after inoculation. The most effective treatments corresponded to the ethanolic extract from *M. depressa* stem bark and the standard α-asarone at 500 µg/mL (Table 5). In the T3, T6, and T9 treatments, a reduction in infection severity (3.4–1.5%) was observed at 250 µg/mL, which is statistically equal to both extracts and α-asarone. In the negative control (T11), 11 days after inoculation, necrosis was observed in the habanero pepper fruits. The 1% Tween 80 solvent (T12) was not toxic to postharvest habanero pepper.

## 3. Discussion

The present contribution is an addition to the in vitro bioprospecting of native plant extracts against the phytopathogenic fungi of habanero pepper [29]. In particular, the *C. lunata* ITC26 was isolated for the first time from the postharvest fruits of habanero pepper [9] with the phytopathogen being recognized as the causal agent of leaf spot, leaf curl, and the subsequent defoliation in *C. frutescens* [7,30], anthracnose in *C. annumm* [31,32], and *C. chinense* seeds in conservation (strain ITCC 02) [33]. In Yucatan, *C. lunata* strains have been isolated from the leaves of *Solanum lycopersicum* L. [26] and *Tridax radiata* Lodd., Ex-Schult., and Schult. f. [27]. Therefore, the contribution of alternatives to the control of *C. lunata* is necessary. The 40 species under study were collected from six different locations and evaluated against *Fusarium equiseti* strain FCHE and *F. oxysporum* strain FCHJ as phytopathogens on habanero pepper [29], the root-knot nematode *Meloidogyne incognita*, *M. javanica* [34], as well as the repellent and oviposition inhibitory effect against *Bemisia tabaci* [35].

The in vitro antifungal assay of 184 extracts from the different vegetative parts obtained of the plant species from the Yucatan Peninsula led to the detection of 13 extracts (7% of the total, Table 2) with activity against *C. lunata* ITC26. The data reveal that *C. lunata* ITC26 is more sensitive to the ethanolic extracts from *A. amorphoides*, *Helicteres baruensis*, *Licaria* sp., *M. depressa*, and *P. neesianum* than the aqueous extracts. This effect is similar to those previously reported with *F. equiseti* FCHE and *F. oxysporum* FCHJ, which were more sensitive (MGI = 100%) to ethanolic extracts from *M. depressa*, *Parathesis cubana,* and *P. neesianum* at 2000 µg/mL [29]. The active aqueous extracts (5.4%) showed a low antifungal capacity (MGI = 25%) at the 3% *w*/*v*. To date, the extracts of the species studied have not been reported against the pathogen *C. lunata* except for aqueous extracts from *M. depressa* stem bark and root bark and *P. neesianum* leaves with the highest inhibition of sporulation and the germination of conidia (100%) at 3% *w*/*v* strain ITC22 isolated from tomato [26]. Other reports of effective aqueous extracts applied in vitro against *C. lunata* include the leaves of *Lawsonia inermis* L. (2%, *w*/*v*) [36], *Ocimum sanctum* (10% *w*/*v*) [37], and *B. flammea* stem bark (3%, *w*/*v*) [27], which showed 21, 75, and 89%, respectively, effect on the mycelial growth of the pathogen.

In contrast, the ethanolic extracts from *M. depressa* bark from stems and roots and *P. neesianum* leaves were lethal against *C. lunata* ITC26. A previous study with the ethanolic extracts of *Calycopteris floribunda* leaves and methanolic extract of *Tribulus terrestris* stem against *C. lunata* showed MICs of 250 and 300 µg/mL, respectively [38,39]. The hexane and chloroformic extracts of *Costus speciosus* rhizome and the chloroformic extract of *Piper betle* presented higher MICs (500–100 µg/mL) [40,41], and the ethanolic extract from *Acorus calamus* leaves inhibited 57% of *C. lunata* growth at 1000 µg/mL [42].

The fractions of the ethanolic extract of *M. depressa* stem bark were less effective against *C. lunata*, showing higher MIC, IC_50_, and IC_95_, as well as the pure α-asarone compound. Previously, it was documented that α-asarone is the major metabolite of the acetonitrile fraction of *M. depressa* stem bark. This is contrary to what was observed against *F. equiseti* and *F. oxysporum,* where α-asarone showed lower IC_50_ (236 and 482 µg/mL, respectively) compared to the ethanolic extract of *M. depressa* bark (IC_50_ = 468 and 944 µg/mL, respectively) and its acetonitrile fraction (IC_50_ = 462 and 472 µg/mL, respectively) [29]. The loss of activity of the extract when fractionated may be due to the loss of material during the fractionation, degradation, and evaporation of the more potent components; it may also be attributed to the loss of the synergistic effect between the compounds in the mixture when they are separated [43]. The metabolite α-asarone from *M. depressa* stem bark could be synergized with other components of the ethanolic extract responsible for the antifungal activity against *C. lunata*. The synergistic effect among compounds has been previously documented; for example, the combination of eugenol and citral showed synergistic antifungal activity against *Penicillium roqueforti*, reducing the dose of oil required to damage the fungal cell membrane [44].

Based on the results of the in vitro antifungal bioassays against *C. lunata* in this study and the most effective and renewable plant part, the ethanolic extracts from *M. depressa* stem bark, *P. neesianum* leaves, as well as α-asarone were selected for evaluation on the postharvest fruits *of C. chinense*. The ethanolic extract from *M. depressa* stem bark was the most effective followed by *P. neesianum*. This study is the first in vivo report of the antifungal potential of *M. depressa*, *P. neesianum*, and α-asarone to control necrosis caused by *C. lunata* on *C. chinense* fruits. The present contribution adds to the few in vivo studies reporting the activity of plant extracts against *C. lunata.* Other reports include the aqueous extract of *B. flammea* stem bark (3%, *w*/*v*), which showed 100% control of the severity of the *C. lunata* infection in the postharvest fruits of *C. chinense* [9], and *Azadirachta indica* oil (1%, *w*/*v*), which controlled 50% of *C. lunata* infection in *Capsicum annuum* [8]. The effect of active ethanolic extracts (2000 µg/mL) and α-asarone (500 µg/mL) on *C. lunata* mycelium and conidia observed with SEM was similar to that reported by Cruz-Cerino et al. [29] against *F. equiseti* and *F. oxysporum*. Ethanolic extracts of *M. depressa* and α-asarone, by contact, deform and dehydrate conidial and fungal cells, tear the cell wall, and rupture the fungal cell membrane, which prevent the synthesis of essential components such as ergosterol [45,46]. The α-asarone is a potent inhibitor of the 3-hydroxy-3-methylglutaryl coenzyme A reductase (HMGR) that catalyzes ergosterol synthesis in fungi [47]. Their synthetic analogs showed a high effect on recombinant HMGR from *Candida glabatra* with IC_50_ values of 42.65 and 28.77 µM for 2-(2-Methoxy-5-nitro-4-propylphenoxy) acetic acid and 2-(2-Methoxy-4-propylphenoxy) acetic acid, respectively [48]. Additionally, the effect of the compound α-asarone, when evaluated at 500 µg/mL on the hyphal morphology of *C. lunata*, is reported for the first time.

The species *M. depressa* belongs to the Annonaceae family, distributed in Central America as far as Honduras, and is a medicinal plant with reported biological activities mainly in humans, such as hypocholesterolemic, hypoglycemic, cytotoxic, antiproliferative, and antiprotozoal activities [49,50,51] among others, with agricultural applications as an antifungal whose aqueous extracts from the stem bark at 3% (*w*/*v*) caused the inhibition of the sporulation of *A. alternata* ITC24 and *C. lunata* ITC22 (100% on both). This species also inhibited the conidial germination of *F. equiseti* ITC32, *Corynespora cassiicola* ITC23, and *C. lunata* ITC22 (61.3, 80.1, and 100%, respectively), while its root bark extracts inhibited the sporulation of *F. equiseti* ITC32, *C. cassiicola* ITC23, and *C. lunata* ITC22 (61.3, 80.1, and 100%, respectively). It also inhibited the sporulation of *C. lunata* ITC22 [26], and *F. equiseti* FCHE, *F. oxysporum* FCHJ, and *P. oxalicum* were isolated from habanero pepper (MIC = 250–1000 µg/mL) [29,52,53]. Finally, it is a growth inhibitor of *Amaranthus hypochondriacus* and *Echinochloa crusgalli* (IC_50_ = 134–457 µg/mL) [53] and a repellent of the phytophagous insect *B. tabaci* [35]; however, it does not have an effect on the root-knot nematodes *M. javanica* and *M. incognita* [34].

The α-asarone is a phenylpropanoid previously isolated from *A. calamus* (leaves and rhizome), *A. gramineus* (rhizome), *Eusideroxylon zwageri* (seeds), *Perilla frutescens* (leaves, stem, and seeds), and *Sphallerocarpus gracilis*, sometimes together with β- and γ-asarone [54]. The present contribution enriches the spectrum of the antifungal activity of α-asarone against fungal phytopathogens, previously reported against *A.a alternata*, *C. lunata*, and *Macrophomina phaseolina* with a lethal effect (growth inhibition = 100%) [52]; *Botrytis cinerea*, *F. oxysporum,* and *Phomopsis obscurans* (GI = 57,7, 43,6, and 41,5%, respectively) to 300 µM [55,56]; as well as *F. equiseti* and *F. oxysporum* (IC_50_ = 236 and 482 µg/mL) [29]. Also, the α-asarone has pesticidal properties as an antifeedant against *Manduca sexta, Heliothis virescens*, and *Helicovarpa zea*; insecticide on *Aedes aegypti* and *Lucila sericata*; and nematocidal against *Caenorhabditis elegans*, *Panagrellus redivivus*, and *Nyppostrongylus brasiliensis* [57,58]. Other compounds reported from *M. depressa* include asaraldehyde, isomyristicin, isoelemicin, 1,2,3,4-tetramethoxy-5-(2-propenyl)-benzene 2,3,4,5-tetramethoxybenzaldehyde, 2,3,4,5-tetramethoxycinnamaldehyde, and 2,3,4,5-tetramethoxycinnamyl alcohol [29,56,59]. Among these compounds, 1,2,3,4-tetramethoxy-5-(2-propenyl)-benzene was the most abundant component of the chloroform extract from *M. depressa* (syn. *Malmea depressa*) as well as in our ethanolic and acetonitrile fraction from the root bark and had antifungal activity on a strain of *F. oxysporum* with a MIC of 250 µg/mL [56]; however, it had no effect on *F. equiseti* FCHE and *F. oxysporum* FCHJ [29], which may be effective against *C. lunata*.

In addition, the species *P. neesianum* confirmed its antifungal capacity, whose ethanolic extract was highly effective in vitro and in the postharvest fruits of habanero pepper to prevent the infection of *C. lunata* ITC26 strain isolated from tomato, while aqueous extract from *P. neesianum* leaves effectively inhibited the sporulation and germination of the conidia of *C. cassiicola* ITC22 isolated from tomato [26]. As previously reported, the ethanol extract and its acetonitrile fraction showed an IC_50_ of 788 and 462 µg/mL on *F. equiseti* FCHE isolated from habanero pepper [29] (Cruz-Cerino et al., 2020). Other studies have previously reported the antifungal effect of *Piper caninum* leaf extract at 3% on *Nigrospora orizae* and *Curvularia verriculosa* pathogens on *Oryza sativa* [60,61]. The essential oils from *P. neesianum* leaves were identified as bicyclogermacrene, germacrene D, and β-caryopyllene (7.5%) and were major compounds among the 19 detected with gas chromatography coupled to mass spectrometry analysis [62].

## 4. Materials and Methods

### 4.1. Plant Collection and Processing

The 40 plants from 6 different sites in the Yucatan Peninsula, Mexico, were collected and processed as previously described by Cruz-Cerino et al. [29]. One specimen of each plant species was deposited in the herbarium U Najil Tikin Xiw (the house of dry grass in Mayan) of the Natural Resources Unit of the Centro de Investigación Científica de Yucatán, A.C. (CICY) with their respective collection numbers (Table 5).

### 4.2. Plant Extracts and Partition of Active Extracts

The aqueous and ethanol extracts were obtained as previously reported by Cruz-Cerino et al. [29]. Briefly, plant material (1.5 g) was extracted for 15 min with boiled water, filtered, and diluted with distilled water (25 mL) to a final concentration of 6% (*w*/*v*). The aqueous extract was sterilized through a 0.22 μm Millipore filter (Merck-Millipore, Burlington, MA, USA) and frozen at −17.5 ± 0.5 °C until use. In comparison, the ethanolic extracts were obtained using sonication at 20 kHz (Cole-Parmer, Chicago, IL, USA) at room temperature for 20 min each time (three times). The solvent was removed under vacuum in a rotary evaporator (IKA model RV-10, Staufen, DE) until dryness was reached.

The ethanol extracts with a lethal effect on *C. lunata* were fractionated with hexane-acetonitrile three times (2:1, 1:1, 1:1 *v*/*v*), obtaining a hexane fraction (a), acetonitrile fraction (b), and methanol-soluble precipitate (c) of each sample after eliminating the solvents via evaporation as described above.

### 4.3. Fungal Cultures

*Curvularia lunata* ITC26 was obtained from the fungal collection of the Phytopathology Laboratory, Tecnológico Nacional de México, campus Conkal. The strain was isolated from lesions of the habanero pepper fruit [9], maintained in (a) 20% glycerol (*v*/*v*) frozen at −80 °C, (b) sterile distilled water, and (c) potato dextrose agar in slant tubes (PDA, BD, Bioxon, Edo. Mex., MX) at 4 °C in the dark.

### 4.4. Antifungal Microdilution Assay of Extracts

#### 4.4.1. Preparation of Conidial Suspension

*C. lunata* strain was cultured on an oat agar culture medium and incubated as described with slight modifications by Cruz-Cerino et al. [29]. Briefly, a sterile saline solution (5 mL) was added to the surface of the fungal culture (11 days), and the conidia were scraped with a sterile brush. Then, the conidial suspension was filtered through a double layer of sterile cheesecloth and adjusted to a final concentration of 1 × 10^5^ conidia/mL using a hemocytometer. The antifungal evaluation of the aqueous and ethanolic extracts against *C. lunata* was carried out with the microdilution bioassay [29].

#### 4.4.2. Bioassay with Aqueous Extracts

The mycelial growth inhibition (MGI) of the *C. lunata* strain was determined using a 96-microwell plate, as described by Abou et al. [63] and, with slight modifications, Cruz-Cerino et al. [29]. The aqueous extracts (100 µL of each 6%) were transferred to each microwell. The fungicide prochloraz (5 µL, 450 g a.i./L; Bayer CropScience, Clayton, NC, USA) and 100 µL of the conidial suspension as the negative control were used. Finally, 100 µL of the conidial suspension was added to each well for a final concentration of 3% *w*/*v* of the aqueous extracts, 0.112% of prochloraz (*w*/*v*), and 5 × 10^4^ conidia/mL of *C. lunata* strain. Each sample was tested in triplicate, and all microdilution plates were maintained at 27 ± 2 °C for 16 h light/8 h dark. The mycelial growth was recorded at 96 h using a 0–4 scale, where 4 is full (0% MGI), and 0 is the absence of MG (MGI = 100%) [64]. The mycelial growth (MG) data were converted to a percentage of MGI using Abbott’s formula: [(% MG in the negative control − % MG in the treatment)/% MG in the negative control)] × 100.

#### 4.4.3. Bioassays with Ethanolic Extracts

The samples (40 µg/µL) were dissolved in a mixture of dimethylsulfoxide (DMSO) (Sigma-Aldrich, St. Louis, MO, USA) with 0.5% Tween 20. RPMI (Roswell Park Memorial Institute 1640) liquid medium (90 µL) was transferred to each microwell; the ethanol extract (10 µL) and 100 µL of the conidial suspension were added to reach a final ethanolic extract concentration of 2000 µg/mL (Merck Millipore Darmstadt, DE). The negative controls were RPMI (Merck Millipore Darmstadt, DE) (100 µL) and water (100 µL), and a mixture of solvents (0.5% Tween-20 DMSO: RPMI 1:9, *v*/*v*) were used. Prochloraz (5 µL) was the positive control, which is described above [29]. Each sample was performed in triplicate, and all the plates were incubated and assessed as described above.

#### 4.4.4. Minimum Inhibitory Concentration

Serial dilutions of the selected ethanol extracts (2000, 1000, 500, 250, and 125 µg/mL) and their fractions (1000, 500, 250, and 125 µg/mL) were prepared as described above and tested using a microdilution assay to determine the MIC [28]. The pure α-asarone standard was acquired from Sigma-Aldrich (St. Louis, MO, USA) and was tested at 500, 250, and 125 µg/mL. The samples were performed with four replicates three times. The same controls and incubation conditions (96 h) were used as described above. The lowest extract concentration, at which no mycelial growth was observed in the wells, was registered as the MIC.

Finally, the fungicidal or fungistatic effect was determined for all microwells that did not grow after 96 h. Then, 10 µL from each microwell was transferred to PDA in a Petri dish and maintained at 27 ± 2 °C. The absence of mycelial growth after 72 h indicated the sample’s fungicidal effect, and the mycelial growth was fungistatic [65].

### 4.5. Evaluation of Ethanolic Extracts on Hyphal Morphology of Curvularia lunata ITC26

The effect of the root and stem bark of *M. depressa* (2000 µg/mL) and α-asarone (500 µg/mL) on the mycelium and conidia of *C. lunata* was observed in a JSM 6360 SEM (Jeol, Tokyo, Japan) at 20 kV. The samples were prepared as previously described by Cruz-Cerino et al. [29]. Briefly, a disk (5 mm) of *C. lunata* grown on oat agar culture medium for 11 days was exposed to 200 µL of ethanol extract. After 96 h, the sample was filtered through a nylon membrane and fixed in a mixture of 2.5% *v*/*v* glutaraldehyde and 0.2 M sodium phosphate (pH 7.2 at 4 °C). After 48 h, the sample was washed (2×, 1 h each time) with phosphate buffer and dehydrated in an ethanol series (1 h each: 30–100%, 2× absolute ethanol). The samples were dried with CO_2_, attached to a sample holder, and coated with gold for 10 min in an ionizing chamber (Dentom Vacuum-Desk II, Moorestown, NJ, USA).

### 4.6. In Vivo Evaluation of Ethanolic Extracts and α-Asarona against C. lunata on Habanero Pepper Fruits

The habanero pepper fruits used in the bioassay were from cv. Jaguar orange when ripe (Seminis Vegetable Seeds^®^, Sant Louis, MO, USA). The habanero pepper fruits were selected according to color and size, discarding those which were wrinkled and damaged. These were washed with tap water and superficially disinfected via immersion in a 70% alcohol solution (1 min) and 2% sodium hypochlorite (1 min), and then double rinsed in sterile distilled water (2 min). Immediately, small lesions were made on the surface of the fruits with the help of a sterile needle to promote infection. All ethanolic extracts were diluted in 1% Tween 80 (Sigma-Aldrich). The treatments (T) evaluated corresponded to the stem bark of *M. depressa* at concentrations of 750 (T1), 500 (T2), and 250 (T3) µg/mL; the leaves of *P. neesianum* at 750 (T4), 500 (T5), and 250 (T6) µg/mL; the α-asarone standard at 500 (T7), 250 (T8), and 125 (T9) µg/mL; the fungicide Mirage^®^ CE45 prochloraz (T10) at a concentration of 450 µg/mL as a positive control; and fruits treated with water (T11) and with 1% Tween 80 (T12) as negative controls. The habanero pepper fruits were submerged individually for 5 min in the ethanolic extracts at the indicated concentrations for each treatment. The fruits were left to dry in a laminar flow hood for 40 min and were inoculated via spraying with a suspension of *C. lunata* spores (1 × 10^5^ spores/mL). At 48 h, it was inoculated for the second time [9]. Each treatment was carried out with five replicates, and they were incubated in plastic trays at a temperature of 27 ± 2 °C until the negative controls showed symptoms of the disease.

At 11 days after inoculation, the severity of the damage in the fruits was estimated, and the percentage of the effectiveness of the extracts was evaluated with the use of a scale of four classes, no visible damage (0), low severity (1), medium severity (2), and severe damage (3), as well as reporting the percentage of the average values (% severity) [9]. The experiment was repeated in 3 different events with 15 replicates per treatment (*n* = 15).

### 4.7. Statistical Analyses

A one-way analysis of variance was performed with the prior transformation of the original % MGI and severity fruits data using the formula y = arsin [sqrt (y/100)]. The treatment means were compared using Tukey’s multiple range test (*p* = 0.05). Variance analyses were performed using SAS ver. 9.4 for Windows (SAS Institute, Cary, NC, USA). Using probit analysis, the IC_50_ and IC_95_ values for the extracts and effective fractions were calculated (95% confidence intervals).

## 5. Conclusions

This research is the first contribution to the in vitro and in vivo evaluations of native plant extracts against the pathogen *C. lunata* associated with the habanero pepper fruit in the Yucatan Peninsula. Our knowledge about antifungal extracts from native Yucatan species was enriched, particularly the spectrum of action of *M. depressa* and *P. neesianum*. The phytopathogen *C. lunata* ITC26 shows greater sensitivity to ethanol extracts of *M depressa* and *P. neesianum* than pure α-asarone. Our native species *M. depressa* and *P. neesianum* are viable alternatives in developing a natural antifungal agent to reduce the severity of *C. lunata* on the postharvest fruits of habanero pepper.

## Figures and Tables

**Figure 1 plants-12-02908-f001:**
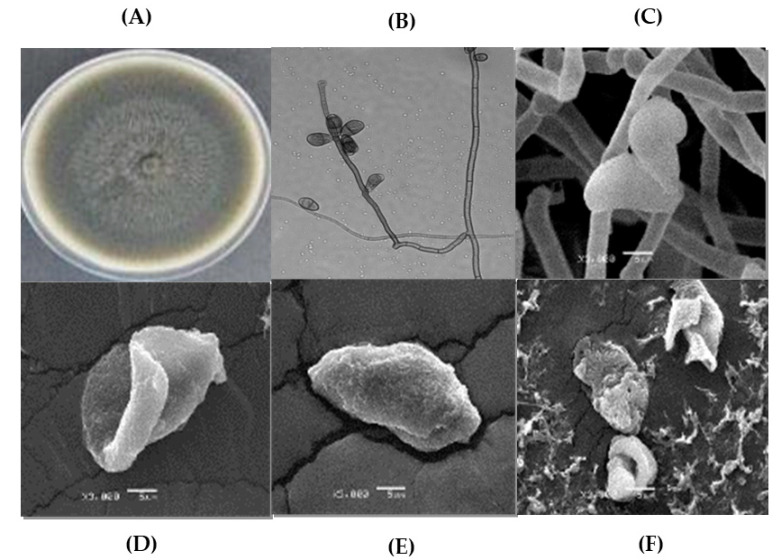
Morphology of *Curvularia lunata* ITC26. (**A**) After seven days on potato dextrose agar; (**B**) conidiophore and conidia of *C. lunata* 1000×; (**C**) micrograph of well-formed mycelium and microconidia with normal growth (negative control); (**D**) conidia contorted by the effect of ethanolic extract from *Mosannona depressa* stem bark at 2000 µg/mL; (**E**) conidia dehydrated by the effect of ethanolic extract from *M. depressa* root bark at 2000 µg/mL; (**F**) collapsed conidia after 96 h exposed to α-asarone at 500 µg/mL.

**Figure 2 plants-12-02908-f002:**
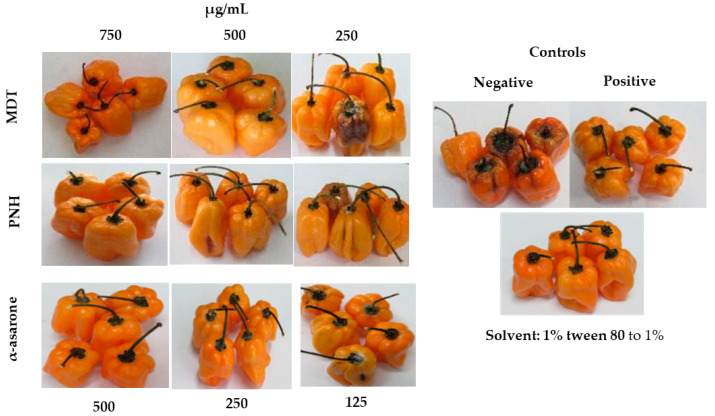
In vivo effectiveness of extracts from *Mosannona depressa* stem bark (MDT), *Piper neesianum* leaves (PNH), and α-asarone against *Curvularia lunata* in habanero pepper fruits. Negative control: conidial suspension; positive control: prochloraz 450 µg/mL.

**Table 1 plants-12-02908-t001:** Percentage of mycelial growth inhibition of *Curvularia lunata* ITC26 by extracts from native species of the Yucatan Peninsula in the microdilution assay.

Plant Species	Concentration	Mycelial Growth Inhibition (%)			
		*Curvularia lunata* ITC26			
		L	S	R	C
	Ethanol (µg/mL)				
*Alvaradoa amorphoides*	2000	75 ^b^	0 ^c^	0 ^d^	
*Helicteres baruensis*	2000	75 ^b^	25 ^b^	0 ^d^	
*Licaria* sp.	2000	0 ^d^	0 ^c^	75 ^b^	
*Mosannona depressa*	2000	0 ^d^	100 ^a^	100 ^a^	
*Piper neesianum*	2000	100 ^a^	0 ^c^	75 ^b^	
	Aqueous (% *w*/*v*)				
*Byrsonima bucidifolia*	3	25 ^c^	25 ^b^	25 ^c^	
*Morella cerifera*	3	25 ^c^	0 ^c^	25 ^c^	
RPMI	Negative C				0 ^b^
blank					0 ^b^
Prochloraz 0.11%	Positive C				100 ^a^

C: control; L: leaves; S: stem; R: root; RPMI: Roswell Park Memorial Institute medium; ne: not evaluated; blank: dimethyl sulfoxide with 0.5% Tween 20. ^a,b,c,d^: means with different letters within columns differ significantly (Tukey’s test, *p* < 0.05). Extracts from *M. depressa* were from the bark from stems and roots.

**Table 2 plants-12-02908-t002:** Minimum inhibitory concentration (MIC) of *Mosannona depressa* extracts, *Piper neesianum,* and α-asarone on *Curvularia lunata* ITC26 in the microdilution assay.

Solvent	Extract/ Fraction	*Curvularia lunata*				
		Concentration (µg/mL)				
		1000	500	250	125	MIC
Ethanol	MDT	100 ^a^	100 ^a^	100 ^a^	0 ^c^	250++
Hexane	MDT-a	50 ^b^	0 ^e^	0 ^g^	0 ^c^	>1000
Acetonitrile	MDT-b	100 ^a^	75 ^c^	25 ^f^	0 ^c^	1000+
Precipitate	MDT-c	100 ^a^	50 ^d^	0 ^c^	0 ^c^	1000+
Ethanol	MDR	100 ^a^	100 ^a^	100 ^a^	0 ^c^	250++
Hexane	MDR-a	50 ^b^	25 ^d^	0 ^g^	0 ^c^	>1000
Acetonitrile	MDR-b	100 ^a^	83 ^b^	50 ^e^	0 ^c^	1000+
Precipitate	MDR-c	100 ^a^	100 ^a^	83 ^c^	0 ^c^	500+
Ethanol	PNH	100 ^a^	100 ^a^	0 ^g^	0 ^c^	500+
Hexane	PNH-a	25 ^c^	25 ^d^	0 ^g^	0 ^c^	>1000
Acetonitrile	PNH-b	100 ^a^	100 ^a^	91 ^b^	0 ^c^	500+
Precipitate	PNH-c	100 ^a^	0 ^e^	0 ^g^	0 ^c^	1000+
CS	α-Asarone	ne	100 ^a^	75 ^d^	25 ^b^	500+
	NC	0 ^d^	0 ^e^	0 ^g^	0 ^c^	
	PC	100 ^a^	100 ^a^	100 ^a^	100 ^a^	

MDT: *Mosannona depressa* stem bark; MDR-b: *M. depressa* root bark; PNH: *Piper neesianum* leaves; NC: negative control (conidial suspension/RPMI: Roswell Park Memorial Institute medium); PC: positive control (prochloraz 0.11%); CS: commercial standard; ne: not evaluated; (+): fungistatic; (++): fungicidal; ^a,b,c,d,e,f,g^: means with different letters within columns differ significantly (Tukey’s test, *p* < 0.05).

**Table 3 plants-12-02908-t003:** IC_50_ and IC_95_ of active extracts and fractions from *Mosannona depressa*, *Piper neesianum*, and α-asarone against mycelial growth of *Curvularia lunata* ITC26.

Source		*Curvularia lunata*	
	Extract/Fraction	IC_50_ (CI)	IC_95_ (CI)
*M. depressa*	MDT	188 (42–308)	218 (81–342)
	MDT-b	388 (298–528)	627 (449–1022)
	MDT-c	388 (298–528)	627 (449–1022)
	MDR	188 (42–308)	218 (81–342)
	MDR-c	229 (210–450)	265 (241–465)
α-asarone	CS	190 (130–271)	325 (253–631)
*P. neesianum*	PNH	378 (278–490)	428 (350–600)
	PNH-b	222 (205–470)	256 (205–404)

CI: confidence interval; CS: commercial standard; MDT: *Mosannona depressa* (stem bark); PNH: *Piper neesianum* leaves; b: acetonitrile fraction; c: precipitate.

**Table 4 plants-12-02908-t004:** Effectiveness of ethanolic extracts of *Mosannona depressa*, *Piper neesianum*, and α-asarone in controlling *Curvularia lunata* infection in habanero pepper fruits after 11 days of exposure.

Extract	Treatment	Concentration µg/mL	Severity (%)	Effectiveness (%)
MDT	T1	750	0 ^e^	100 ^a^
	T2	500	0 ^e^	100 ^a^
	T3	250	3.4 ^cd^	93.2 ^bcd^
PNH	T4	750	0 ^e^	100 ^a^
	T5	500	1.5 ^d^	97 ^bc^
	T6	250	8 ^cb^	84 ^c^
α-asarona	T7	500	0 ^e^	100 ^a^
	T8	250	2.9 ^d^	94.2 ^bc^
	T9	125	8 ^cb^	84 ^c^
PC	T10	450	0 ^e^	100 ^a^
NC	T11	-	100 ^a^	0 ^e^
solvent	T12	-	0 ^e^	0 ^e^

% Effectiveness: average of five replicates. ^a,b,c,d,e^: means with different letters within the columns differ significantly (Tukey’s test, *p* < 0.05). NC: negative control (conidial suspension); PC: positive control (prochloraz 450 µg/mL); solvent: Tween 80 al 1%. MDT: *Mosannona depressa* stem bark; PNH: *Piper neesianum* leaves.

**Table 5 plants-12-02908-t005:** Native plants from the Yucatán Peninsula tested against *Curvularia lunata* ITC26.

Family	Species	Site	Voucher	Plant Parts
Rubiaceae	*Alseis yucatanensis* Standl.	1	JLT-3179	L
Simaroubaceae	*Alvaradoa amorphoides* Liebm.	2	GC-8236	L, S, R
Annonaceae	*Annona primigenia* Standl. and Steyerm	2	GC-8057	L, SB
Malvaceae	*Bakeridesia notolophium* (A. Gray) Hochr.	3	RD-s/n	L, S
Acanthaceae	*Bravaisia berlandieriana* (Nees) T.F.Daniel	4	GC-8168	L, S, R
Malpighiaceae	*Byrsonima bucidifolia* Standl.	2	GC-8087	L, S, R
Asteraceae	*Calea jamaicensis* (L.) L.	2	GC-8084	WP
Apocynaceae	*Cameraria latifolia* L.	2	JLT-1165	L, SB, R
Sapotaceae	*Chrysophyllum mexicanum* Brandegee ex Standl.	2	GC-8082	L, S, R
Polygonaceae	*Coccoloba* sp.	5	GC-8258	L, S
Euphorbiaceae	*Croton arboreus* Millsp.	2	JLT-1132	L, S, R
Euphorbiaceae	*Croton itzaeus* Lundell	2	JLT-1138	L, SB, RB
Euphorbiaceae	*Croton* sp.	5	GC-8262	WP
Sapindaceae	*Cupania* sp.	6	GC-8009	L, S
Ebenaceae	*Diospyros* sp.	4	GC-8147	L
Erythroxylaceae	*Erythroxylum confusum* Britton	2	JLT-1143	L, S, R
Erythroxylaceae	*Erythroxylum rotundifolium* Lunan	2	GC-8179	L, S
Erythroxylaceae	*Erythroxylum* sp.	4	GC-8137	L
Myrtaceae	*Eugenia* sp.	4	GC-8127	L, S, R
Euphorbiaceae	*Euphorbia armourii* Millsp.	1	JLT-3182	WP
Rubiaceae	*Guettarda combsii* Urb.	2	GC-8047	L, SB, RB
Malvaceae	*Helicteres baruensis* Jacq.	1	GC-8127	L, S, R
Malpighiaceae	*Heteropterys laurifolia* (L.) A. Juss.	2	GC-8035	L, SB, R
Violaceae	*Hybanthus yucatanensis* Millsp.	4	GC-8158	L, S
Convolvulaceae	*Ipomoea clavata* (G. Don) Ooststr. Ex J.F.Macbr.	1	JLT-3181	WP
Rhamnaceae	*Karwinskia humboldtiana* (Willd. Ex Roem. and Schult.) Zucc.	1	JLT-3188	L
Lauraceae	*Licaria* sp.	2	GC-8037	L, SB, RB
Apocynaceae	*Macroscepis diademata* (Ker Gawl.) W.D. Stevens	1	JLT-3187	L, SB
Malpighiaceae	*Malpighia glabra* L.	4	GC-8144	L, S, R
Myricaceae	*Morella cerifera* (L.) Small.	2	JLT-1137	L, S, RB
Annonaceae	*Mosannona depressa* (Ball.) Chatrou	2	GC-8085	L, SB, RB
Primulaceae	*Parathesis cubana* (A. DC.) Molinet and M.Gómez	2	JLT-1133	L, SB, RB
Sapindaceae	*Paullinia* sp.	4	GC-8106	L, R
Piperaceae	*Piper neesianum* C.DC.	2	GC-8080	L, S, R
Rubiaceae	*Psychotria nervosa* Sw.	2	GC-8086	WP
Rubiaceae	*Randia aculeata* L.	4	GC-8156	L, S, R
Sapindaceae	*Serjania caracasana* (Jacq.) Willd	4	GC-8114	L, S, R
Simaroubaceae	*Simarouba glauca* DC.	2	GC-8081	L, SB, RB
Apocynaceae	*Stemmadenia donnell-smithii* (Rose) Woodson	2	GC-8056	L, SB
Passifloraceae	*Turnera aromatica* Arbo	2	GC-8081	WP

L: leaves; RB: root bark; S: stem; SB: stem bark; R: root; WP: whole plant; site 1: Kaxil Kiuic; site 2: Jahuactal; site 3: Punta Pulticub; site 4: Punta Laguna; site 5: Xmaben; site 6: Chacchoben Limones.

## Data Availability

The data are contained in the article and in the Appendix A, and are also accessible in Mexico’s national repository (http://cicy.repositorioinstitucional.mx/jspui/handle/1003/2688, accessed on 5 August 2023).

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
