# Peer review of "Plant Extracts from the Yucatan Peninsula in the In Vitro Control of Curvularia lunata and Antifungal Effect of Mosannona depressa and Piper neesianum Extracts on Postharvest Fruits of Habanero Pepper"

_plants, 2023, doi:10.3390/plants12162908_

Round 1

Reviewer 1 Report

This work represents the continuity of previous work carried out by the group. In this sense, the work is very descriptive and I consider that it should provide more relevant information to the reader, not only a list of analyzed plants. 

The title should be modified to a more specific one.

What is the metabolite composition of the plants that present activity?

What is the mechanism of action of the extracts?

Table 1 included in the introduction should be included in results or in materials and methods.

What is the application of the fractions in acetonitrile or hexane?

Author Response

Comments and Suggestions for Authors

Thank you for your valuable and detailed observations. Your input has improved the manuscript.

Modifications you recommended have mostly been included in the manuscript, with recommendations of Reviewer 2 (red: eliminated, blue: added).

This work represents the continuity of previous work carried out by the group. In this sense, the work is very descriptive and I consider that it should provide more relevant information to the reader, not only a list of analyzed plants. 

The title should be modified to a more specific one.

R= The title was modified:

Plant extracts from the Yucatan Peninsula in the in vitro control of Curvularia lunata and antifungal effect of Mosannona depresa and Piper neesianum extracts on postharvest fruits of habanero pepper

What is the metabolite composition of the plants that present activity?

R: Lines 291 – 294: added:

Other compounds reported from M. depressa included asaraldehyde, isomyristicin, isoelemicin, 1,2,3,4-tetramethoxy-5-(2-propenyl)-benzene 2,3,4,5-tetramethoxybenzaldehyde, 2,3,4,5-tetramethoxycinnamaldehyde, and 2,3,4,5-tetramethoxycinnamyl alcohol [29,56,59].

Lines 308-311 added:

The essential oils from P. neesianum leaves were identified bicyclogermacrene, germacrene D and β-caryopyllene (7.5%) as major compounds among 19 detected by gas chromatography coupled to mass spectrometry analysis [62].

What is the mechanism of action of the extracts?

R: Crude plant extracts are a mixture of many compounds that may combine to suppress the growth of phytopathogenic fungi. Plant extracts with varied antifungal compounds may have several mechanisms of action that could reduce the emergence of resistance in pathogens.

Table 1 included in the introduction should be included in results or in materials and methods.

R= Table 1 was moved to materials and methods, now Table 5.

What is the application of the fractions in acetonitrile or hexane?

R= Partitioning with immiscible solvents is the first fractionation of the crude extract that separates the mixture's components by polarity. Low polarity compounds such as fatty acids and monoterpenes are obtained with hexane, and medium and high polarity compounds are obtained with acetonitrile. Biological evaluations are made with these fractions, and we observe synergism or antagonism effects.  This data let us design strategies for the purification process and identify the compounds responsible for the biological activity.

Reviewer 2 Report

In the manuscript, the authors evaluated the in vitro antifungal effect of ethanol and aqueous extracts from different vegetative parts of 40 native plants of the Yucatan Peninsula on Curvularia lunata ITC26, a pathogen of habanero pepper (Capsicum chinense). This work is well presented and the results of this work contained some interesting data. However, the manuscript needs revision and some minor issues still need to be improved.

- There are few explanations of the rationale for the experiment design. No rationale is given as to why these specific plants were chosen.

- There's a discussion of the results, but the relationship between antifungal effects and possible causes is not clear . What was the chemical compositions of the ethanol extracts of Mosannona depressa  ? Is there any other chemical compounds of the extracts else which were taken part in the antifungal activities except α-asarone?

-The 40 native plants of the Yucatan Peninsula were extracted by the same extraction conditions. How are the extraction conditions and the choice of solvent determined? If the extraction conditions will influence the antifungal effect of the extracts?

acceptable

Author Response

In the manuscript, the authors evaluated the in vitro antifungal effect of ethanol and aqueous extracts from different vegetative parts of 40 native plants of the Yucatan Peninsula on Curvularia lunata ITC26, a pathogen of habanero pepper (Capsicum chinense). This work is well presented and the results of this work contained some interesting data. However, the manuscript needs revision and some minor issues still need to be improved.

Thank you for your valuable and detailed observations. Your input has improved the manuscript.

Modifications you recommended have mostly been included in the manuscript, with recommendations of Reviewer 2 (red: eliminated, blue: added).

- There are few explanations of the rationale for the experiment design. No rationale is given as to why these specific plants were chosen.

R: Added in the introduction Lines 72-76 added:

Continuing the bioprospecting search for natural plant agrochemical options to control plant pathogens and parasites, our group collected a second and larger number of species from six sites in the Yucatan peninsula. The criteria for the selection of plant species were local availability, chemotaxonomical and ethnobotanical antecedents, absence of environmental risk restrictions, and some of them at serendipity.

- There's a discussion of the results, but the relationship between antifungal effects and possible causes is not clear .

R: Lines 255 -262 added:

Ethanolic extracts of M. depressa, and α-asarone by contact deforming and dehydrating conidial and fungal cells, tearing the cell wall and rupturing the fungal cell membrane, which prevents the synthesis of essential components such as ergosterol [45, 46]. The α-asarone is a potent inhibitor of the 3-hydroxy-3-methylglutaryl coenzyme A reductase (HMGR) that catalyzes ergosterol synthesis in fungi [47]; and their synthetic analogues showed a high effect on recombinant HMGR from Candida glabatra, with IC50 values of 42.65  and  28.77 µM, for  2-(2-Methoxy-5-nitro-4-propylphenoxy) acetic aci  d and 2-(2-Methoxy-4-propylphenoxy) acetic acid, respectively [48].

What was the chemical compositions of the ethanol extracts of Mosannona depressa  ?  R: Lines 291 – 294 added:

Other compounds reported from M. depressa included asaraldehyde, isomyristicin, isoelemicin, 1,2,3,4-tetramethoxy-5-(2-propenyl)-benzene 2,3,4,5-tetramethoxybenzaldehyde, 2,3,4,5-tetramethoxycinnamaldehyde, and 2,3,4,5-tetramethoxycinnamyl alcohol [29,56,59].

Is there any other chemical compounds of the extracts else which were taken part in the antifungal activities except α-asarone?

R= Yes, there is 2,3,4-tetramethoxy-5-(2propenyl)-benzene.

Lines 294 – 299 added:

Among these compounds, 1,2,3,4-tetramethoxy-5-(2propenyl)-benzene was the most abundant component of the chloroform extract from M. depressa (syn. Malmea depressa), as well as in our ethanolic and acetonitrile fraction from the roots bark, and had antifungal activity on a strain of F. oxysporum with a MIC of 250 µg/mL [56], however had no effect on F. equiseti FCHE and F. oxysporum FCHJ [29] which may be effecitve against C. lunata.

-The 40 native plants of the Yucatan Peninsula were extracted by the same extraction conditions.

  1. All aqueous extracts were obtained by maceration with boiled water, and all ethanol extracts were obtained by sonication. The ethanolic extraction was complemented in the experimental part.

How are the extraction conditions and the choice of solvent determined?

R:  Based on the literature, experience, available material, and the type of metabolites to be extracted, the method and solvent(s) to be used are selected. We use ethanol as a solvent to extract plant material because it is polar and less toxic than methanol. Both solvents are commonly used under various conditions and can extract low, medium and high polarity compounds from cells.  Water extracts more polar hydrophilic compounds than those obtained with ethanol or methanol.

According to the quantities to be extracted, equipment, and material availability, we use maceration for more than one kilo and sonication for grams of material. In both methods, the extraction is good, but in less time with sonication, both at room temperature.

If the extraction conditions will influence the antifungal effect of the extracts?

R=  Sometimes, the solvents and extraction conditions may influence the types of compounds to be extracted and induce artifacts, which are modified compounds not naturally occurring in plants.  These artifacts can lose their activity when their functional groups are modified and cannot bind to their target of action.

Round 2

Reviewer 2 Report

In the manuscript, the authors evaluated the in vitro antifungal effect of ethanol and aqueous extracts from different vegetative parts of 40 native plants of the Yucatan Peninsula on Curvularia lunata ITC26, a pathogen of habanero pepper (Capsicum chinense). This work is well presented and the results of this work contained some interesting data. The Submission has been greatly improved and is worthy of publication. However, some minor issues still need to be improved.

-The site of  figure1 and figure 2 should be adjusted.

-line 408 \line 432 : there should be two empty spaces at the beginning of the paragraph

acceptable

Author Response

Thank you very much for your valuable comments to improve the manuscript.

-The site of  figure1 and figure 2 should be adjusted.

R.­-Figures 1 and 2 were adjusted after lines 143 and 178, respectively.

-line 408 \line 432 : there should be two empty spaces at the beginning of the paragraph

R.- Corrections made

Line 408 now is line 402

line 432 is now line 427